# Simplified cough test can predict the risk for pneumonia in patients with acute stroke

**Masahiro Nakamori**[1,2]*, **Eiji Imamura**[1], **Miyu Kuwabara**[3], **Tomoko Ayukawa**[4],
**Keisuke Tachiyama**[1,2], **Teppei Kamimura**[1,2], **Yuki Hayashi**[1,2], **Hayato Matsushima**[1],
**Mika Funai**[3], **Tatsuya Mizoue**[5], **Shinichi Wakabayashi**[5]

**1** Department of Neurology, Suiseikai Kajikawa Hospital, Hiroshima, Japan, **2** Department of Clinical
Neuroscience and Therapeutics, Hiroshima University Graduate School of Biomedical and Health Sciences,
Hiroshima, Japan, **3** Department of Nursing, Suiseikai Kajikawa Hospital, Hiroshima, Japan, **4** Department of
Rehabilitation, Suiseikai Kajikawa Hospital, Hiroshima, Japan, **5** Department of Neurosurgery, Suiseikai
Kajikawa Hospital, Hiroshima, Japan

* mnakamori1@gmail.com

journal.pone.0239590

System, Affiliated with Icahn School of Medicine at
Mount Sinai, NY, USA, UNITED STATES

**Data Availability Statement:** All relevant data is
within the Supporting Information file.

## Abstract

We investigated the association between the results of a simplified cough test and pneumonia onset in 226 patients with acute stroke admitted to Suiseikai Kajikawa Hospital from April to December, 2018. For the simplified cough test, performed on admission, patients orally inhaled a mist of 1% citric acid–physiological saline using a portable mesh nebulizer. When the first cough was evoked or if it remained absent for 30 seconds (indicating an abnormal result), the test was ended. Patients also completed the repetitive saliva swallowing test (RSST) and modified water swallowing test. We monitored patients for pneumonia signs for 30 days post-admission. Eighteen patients exhibited an abnormal simplified cough test result. On multivariate analysis, an abnormal RSST result was independently associated with an abnormal simplified cough test result. Seventeen patients developed pneumonia. The adjusted Cox proportional hazard model for pneumonia onset revealed that the simplified cough test had predictive power for pneumonia onset (hazard ratio, 10.52; 95% confidence interval, 3.72–29.72). The simplified cough test is a strong indicator for predicting the pneumonia development in patients with acute stroke; it should be added to existing bedside screening tests for predicting pneumonia risk, allowing appropriate and timely intervention.

## Introduction

Aspiration pneumonia is a critical issue in patients with stroke. Prevention of aspiration pneumonia is associated with reduced duration of hospitalization as well as mortality [1–3]. Pneumonia rates have been found to be lower in hospitals that use screening protocols for aspiration in patients with stroke, followed by appropriate interventions for patients identified as being at risk for aspiration pneumonia [4].

Videofluoroscopic and videoendoscopic examinations are considered the most accurate instrumental assessment tools for evaluating aspiration. However, some patients cannot

**Funding:** The authors received no specific funding for this work.

**Competing interests:** The authors have declared that no competing interests exist.

undergo sufficient testing due to their general condition or the unavailability of such tools in some medical institutions. Hence, bedside screening protocols for assessing swallowing dysfunction that are non-invasive and simple to perform are often used, including the repetitive saliva swallowing test (RSST) and modified water swallowing test (MWST) [5, 6]. Additionally, other complementary methods such as tongue pressure measurement and tongue ultrasonography are reportedly useful for evaluating swallowing dysfunction [7–9].

Besides assessing swallowing dysfunction, it is important to evaluate the risk of silent aspiration. Adequate swallowing and cough reflexes are important for preventing aspiration pneumonia [10, 11]. Silent aspiration is caused by the lack of a cough reflex and throat clearing when materials are aspirated into the subglottic area. It has been reported that silent aspiration occurs in about 68% of aspirating patients [12]. Given that many screening tests for aspiration involve assessment of the presence of cough, aspiration without cough is often missed. Indeed, one report noted that approximately 40% of silent aspiration cases are overlooked in bedside screening of swallowing dysfunction [13]. Thus, appropriate methods of determining a patient's risk of silent aspiration are needed.

One approach for detecting the risk of silent aspiration is to use an evoked cough reflex test. The cough reflex can be evoked by several methods using mechanical or chemical stimulation. Among these is a cough test in which patients orally inhale a mist of citric acid–physiological saline by means of a nebulizer [14]. In the original method, 5 coughs within 1 minute were considered normal, but this markedly strained the patients. Sato *et al.* [15] suggested a simplified cough test utilizing 1% citric acid from a portable nebulizer, in which the first cough reflex occurring within 30 seconds was judged as normal. Lee *et al.* [16] validated the usefulness of this simplified cough test using videofluoroscopic examinations, whereas this study was performed among a heterogeneous patient population. There are few reports in which this test was performed in the acute stroke patients. This simplified cough test is an easy-to-implement bedside screening test that is performed within a few minutes while the patient is in a supine position. It requires minimal patient cooperation and is less invasive and less expensive than videofluoroscopic or videoendoscopic examinations.

Based on the above information, we hypothesized that the simplified cough test may be useful for predicting pneumonia onset in patients with acute stroke, which would allow for better patient management and outcomes. In this study, we therefore investigated the association between the results of the simplified cough test and pneumonia onset in patients with acute stroke.

## Materials and methods

### Ethics

The study protocols were approved by the ethics committee of Suiseikai Kajikawa Hospital and were performed according to the guidelines of the national government based on the Helsinki Declaration of 1964 and its later amendments. Written informed consent was obtained from all patients or their relatives. All data analyses were performed in a blinded manner.

### Subjects

Patients with acute stroke who were admitted to Suiseikai Kajikawa Hospital from 1 April 2018 to 31 December 2018 were enrolled in this prospective study. We included patients with ischemic stroke or hemorrhagic stroke, who were admitted within 3 days from onset, were aged $\geq$ 20 years, and for whom consent to participate in this study was obtained from the patient or the patient's relatives. We excluded patients who were not eating food before stroke onset, who were in a coma (best eye response score on the Glasgow coma scale of 1), who had

received an operation or endovascular treatment, or who were on mechanical ventilation. Patients who had a history of neuromuscular disorders or asthma were also excluded.

## Simplified cough test

The simplified cough test was performed on admission. Patients orally inhaled a mist of 1% citric acid–physiological saline using a portable mesh nebulizer (NE-U22, Omron, Kyoto, Japan). Citric acid was purchased from Kenei Pharmaceutical Co., Ltd. (Osaka, Japan). Patients were verbally instructed to inhale deeply the nebulized citric acid through the mouth several times, until the first cough occurred. If patients had difficulty inhaling through the mouth, a nose clip was applied to encourage the patient to inhale orally. When the first cough was evoked or if it remained absent for 30 seconds, the test was ended. If a cough was not evoked within the 30-second time frame, we judged the outcome as abnormal.

## RSST and MWST

Patients also performed the RSST and MWST on admission. In the RSST, patients were instructed to swallow as many times as possible in 30 seconds. Two or less swallows within 30 seconds was considered abnormal [5]. In the MWST, patients were asked to drink 3 mL of water. Their drinking patterns and vocal changes after drinking were recorded and scored as follows: 1, no drinking with choking and/or respiratory distress; 2, drinking and respiratory distress; 3, drinking and choking and/or hoarseness without respiratory distress; 4: drinking without choking and respiratory distress; 5: drinking without choking and respiratory distress, plus able to perform two repetitive dry swallows within 30 seconds. A score < 4 was considered abnormal [6].

## Data acquisition

Stroke subtype was determined according to the Trial of Org 10172 in Acute Stroke Treatment criteria [17]. Two stroke specialists evaluated the stroke severity of the patients on admission. The severity of stroke was evaluated using the National Institutes of Health Stroke Scale (NIHSS) score [18].

Baseline demographic and clinical characteristic data, including age, sex, body mass index, smoking habits, and comorbidities (hypertension, diabetes mellitus, dyslipidemia, atrial fibrillation, and stroke) were collected from all patients. In addition, angiotensin-converting enzyme inhibitor, beta blocker, and cilostazol (antiplatelet drug inhibiting phosphodiesterase), which promote cough reflex, were administered [19–21].

We monitored patients for signs of pneumonia during the 30 days following admission. If patients were discharged before day 30, we monitored the patients until the day of discharge. Clinically defined pneumonia was diagnosed based on the agreement of two physicians (M.N. and Y.H.), according to the criteria of the Centers for Disease Control and Prevention [22], *i.e.* the presence of a new and persistent infiltrate or consolidation on at least one chest X-ray or computed tomography examination, with one of the following clinical signs: fever, leucopoenia or leukocytosis, and altered mental status in patients > 70 years of age, in the absence of other causes. In addition, two of the following signs had to be present: new-onset purulent sputum or change in the character of the sputum, new-onset or progressive cough, rales, and impaired gas exchange.

## Statistical analysis

Statistical analyses were performed as in a previous report [5]. The data are expressed as the mean ± standard deviation or the median (minimum, maximum) for continuous variables

and frequencies and percentages for discrete variables. Statistical analysis was performed using JMP statistical software version 14.0 (SAS Institute Inc., Cary, NC, USA). The statistical significance of intergroup differences was assessed using the $\chi^2$ test, Mann–Whitney $U$ test, or an unpaired $t$-test, as appropriate. We calculated the required sample size according to past investigations for pneumonia in acute stroke patients [7]. Based on an alpha level = 0.05, and power = 0.80, we estimated that we would require a total of n = 172 participants. First, univariate analyses were performed. Then, multivariate analyses were performed using select factors that achieved a p value of < 0.10 in the univariate analyses. Kaplan–Meier and Cox proportional hazard regression analyses were performed to test the difference in the development of pneumonia between the normal and abnormal simplified cough test groups. In addition, the sensitivities, specificities, positive predictive values, negative predictive values, and accuracies of the three screening tests (RSST, MWST, and simplified cough test) for predicting pneumonia development were determined. We considered p < 0.05 reflective of statistical significance.

## Results

During the study period, 226 patients with acute stroke were included (mean ± standard deviation age: 72.8 ± 13.1 years; 98 women). Patients' demographic and clinical characteristics are shown in Table 1. The number of patients with an abnormal simplified cough test result was 18 (8.0%). Seventeen out of 226 patients developed pneumonia during the observation period. No patient had pneumonia before stroke onset. Univariate analyses comparing patients with normal and abnormal simplified cough test results showed that the NIHSS score (p < 0.001) and number of patients with an abnormal RSST result (p < 0.001), abnormal MWST result (p = 0.006), and pneumonia onset (p<0.001) were significantly different between the groups. Multivariate analyses including the factors in Table 1 other than pneumonia onset revealed that an abnormal RSST result was independently associated with an abnormal simplified cough test result (odds ratio, 4.27; 95% confidence interval [CI], 1.09–16.73; p = 0.037). The medications of angiotensin-converting enzyme inhibitor, beta blocker, and cilostazol were not associated with the results of simplified cough test.

Potential factors associated with pneumonia onset were evaluated and are listed in Table 2. On univariate analysis, NIHSS score, abnormal RSST result, abnormal MWST result, and abnormal simplified cough test result were identified as significant factors. To avoid confounding factors, we performed the multivariate analysis using the NIHSS score with the abnormal RSST result (model 1), abnormal MWST result (model 2), and abnormal simplified cough test result (model 3) individually (Table 3). These analyses revealed that the RSST and simplified cough test were significantly associated with pneumonia onset. The medications of angiotensin-converting enzyme inhibitor, beta blocker, and cilostazol were not associated with the results of pneumonia onset.

A Kaplan–Meier analysis was used to compare pneumonia development between patients with normal and abnormal simplified cough test results (Fig 1). The abnormal group had a significantly higher incidence of pneumonia than did the normal group (p < 0.001, log-rank test). A Cox proportional hazard model for pneumonia onset was constructed with the abnormal simplified cough test results and adjustment for the NIHSS score. In this model, an abnormal simplified cough test result was also found to be independently associated with pneumonia onset (hazard ratio, 10.52; 95% CI, 3.72–29.72; p < 0.001). All patients who were diagnosed with pneumonia and had an abnormal simplified cough test result developed pneumonia within 10 days from admission.

We compared the utility of the RSST, MWST, and simplified cough test for predicting pneumonia development in patients with acute stroke. Among the 17 patients who developed

**Table 1. Patient demographic and clinical characteristics.**

| Factors | All | simplified cough test | | p value |
| | | abnormal | normal | |
| | n = 226 | n = 18 | n = 208 | |
|---|---|---|---|---|
| Age, year | 72.8 ± 13.1 | 74.2 ± 15.9 | 72.7 ± 12.9 | 0.633 |
| Gender (Female), n (%) | 98 (43.4) | 8 (44.4) | 90 (43.3) | 0.923 |
| Body mass index, kg/m2 | 23.3 ± 4.0 | 23.6 ± 3.8 | 23.3 ± 4.0 | 0.758 |
| Stroke Subtypes | | | | 0.296 |
| ATBI, n (%) | 38 (16.8) | 4 (22.2) | 34 (16.3) | |
| CEI, n (%) | 23 (10.2) | 2 (11.1) | 21 (10.1) | |
| LI, n (%) | 42 (18.6) | 4 (22.2) | 38 (18.3) | |
| Others, n (%) | 91 (40.3) | 3 (16.7) | 88 (42.3) | |
| ICH, n (%) | 32 (14.1) | 5 (27.8) | 27 (13.0) | |
| Location of lesion | | | | |
| frontal lobe, n (%) | 72 (31.9) | 5 (27.8) | 67 (32.2) | 0.699 |
| temporal lobe, n (%) | 24 (10.6) | 2 (11.1) | 22 (10.6) | 0.944 |
| parietal lobe, n (%) | 34 (15.0) | 2 (11.1) | 32 (15.4) | 0.627 |
| occipital lobe, n (%) | 26 (11.5) | 0 (0) | 26 (12.5) | 0.111 |
| insular cortex, n (%) | 10 (4.4) | 2 (11.1) | 8 (3.9) | 0.151 |
| corona radiata, n (%) | 56 (24.8) | 7 (38.9) | 49 (23.6) | 0.148 |
| basal ganglia, n (%) | 52 (23.0) | 8 (44.4) | 44 (21.2) | 0.024 |
| capsulae internae, n (%) | 9 (4.0) | 2 (11.1) | 7 (3.4) | 0.107 |
| thalamus, n (%) | 24 (10.6) | 2 (11.1) | 22 (10.6) | 0.944 |
| brain stem, n (%) | 44 (19.5) | 2 (11.1) | 42 (20.2) | 0.351 |
| cerebellum, n (%) | 20 (8.9) | 2 (11.1) | 18 (8.7) | 0.725 |
| Past history | | | | |
| Hypertension, n (%) | 150 (66.4) | 12 (66.7) | 138 (66.4) | 0.978 |
| Diabetes mellitus, n (%) | 46 (20.4) | 4 (22.2) | 42 (20.2%) | 0.837 |
| Dyslipidemia, n (%) | 35 (15.5) | 2 (11.1) | 33 (15.9) | 0.593 |
| Atrial fibrillation, n (%) | 20 (8.9) | 1 (5.6) | 19 (9.1) | 0.608 |
| Stroke, n (%) | 63 (27.9) | 4 (22.2) | 59 (28.4) | 0.577 |
| Current smoker, n (%) | 51 (22.6) | 4 (22.2) | 47 (22.6) | 0.971 |
| Habitual drinker, n (%) | 75 (33.2) | 4 (22.2) | 71 (34.1) | 0.303 |
| NIHSS score on admission, median (minimum, maximum) | 2 (0, 23) | 7 (0, 23) | 2 (0, 19) | < 0.001 |
| RSST<3 times /30 seconds, n (%) | 37 (16.4) | 10 (55.6) | 27 (13.0) | < 0.001 |
| MWST score<4, n (%) | 15 (6.6) | 4 (22.2) | 11 (5.3) | 0.006 |
| Medication | | | | |
| angiotensin-converting enzyme inhibitor, n (%) | 21 (9.3) | 3 (16.7) | 18 (8.7) | 0.261 |
| beta blocker, n (%) | 36 (15.9) | 4 (22.2) | 32 (15.4) | 0.447 |
| cilostazol, n (%) | 49 (21.7) | 3 (16.7) | 46 (22.1) | 0.590 |
| pneumonia onset, n (%) | 17 (7.5) | 10 (55.6) | 7 (3.4) | < 0.001 |

SD, standard deviation; ATBI, atherothrombotic brain infarction; CEI, cardiogenic embolism infarction; LI, lacunar infarction; ICH, intracerebral hemorrhage; NIHSS, National Institutes of Health Stroke Scale; RSST, repetitive saliva swallowing test; MWST, modified water swallowing test.

pneumonia, 10 patients had an abnormal simplified cough test result. However, among these 10 patients, 6 patients had a normal RSST or MWST test result. The sensitivities, specificities, positive predictive values, negative predictive values, and accuracies of these tests are shown in

**Table 2. Potential factors associated with pneumonia onset.**

| | pneumonia onset | | |
|---|---|---|---|
| | (+) | (-) | univariate analysis |
| **Factors** | **n = 17** | **n = 209** | **p value** |
| Age, year | 77.4 ± 9.5 | 72.4 ± 13.3 | 0.134 |
| Gender (Female), n (%) | 5 (29.4) | 93 (44.5) | 0.227 |
| Body mass index, kg/m2 | 23.0 ± 4.1 | 23.3 ± 4.0 | 0.790 |
| Stroke Subtypes | | | 0.137 |
| ATBI, n (%) | 2 (11.8) | 36 (17.2) | |
| CEI, n (%) | 1 (5.9) | 22 (10.5) | |
| LI, n (%) | 2 (11.8) | 40 (19.1) | |
| Others, n (%) | 6 (35.3) | 85 (40.7) | |
| ICH, n (%) | 6 (35.3) | 26 (12.4) | |
| Location of lesion | | | |
| frontal lobe, n (%) | 4 (23.5) | 68 (32.5) | 0.443 |
| temporal lobe, n (%) | 3 (17.7) | 21 (10.1) | 0.328 |
| parietal lobe, n (%) | 2 (11.8) | 32 (15.3) | 0.694 |
| occipital lobe, n (%) | 2 (11.8) | 24 (11.5) | 0.972 |
| insular cortex, n (%) | 2 (11.8) | 8 (3.8) | 0.126 |
| corona radiata, n (%) | 5 (29.4) | 51 (24.4) | 0.645 |
| basal ganglia, n (%) | 8 (47.1) | 44 (21.1) | 0.014 |
| capsulae internae, n (%) | 0 (0) | 9 (4.3) | 0.383 |
| thalamus, n (%) | 3 (17.7) | 21 (10.1) | 0.328 |
| brain stem, n (%) | 2 (11.8) | 42 (20.1) | 0.404 |
| cerebellum, n (%) | 2 (11.8) | 18 (8.6) | 0.660 |
| Past history | | | |
| Hypertension, n (%) | 10 (58.8) | 140 (67.0) | 0.493 |
| Diabetes mellitus, n (%) | 4 (23.5) | 42 (20.1) | 0.735 |
| Dyslipidemia, n (%) | 3 (17.7) | 32 (15.3) | 0.798 |
| Atrial fibrillation, n (%) | 1 (5.9) | 19 (9.1) | 0.654 |
| Stroke, n (%) | 6 (35.3) | 57 (27.3) | 0.478 |
| Current smoker, n (%) | 2 (11.8) | 49 (23.4) | 0.268 |
| Habitual drinker, n (%) | 1 (5.9) | 74 (35.4) | 0.013 |
| NIHSS score, median (minimum, maximum) | 10 (0, 23) | 2 (0, 19) | <0.001 |
| RSST<3 times /30 seconds, n (%) | 11 (64.7) | 26 (12.4) | <0.001 |
| MWST score<4, n (%) | 5 (29.4) | 10 (4.8) | <0.001 |
| Medication | | | |
| angiotensin-converting enzyme inhibitor, n (%) | 3 (17.7) | 18 (8.6) | 0.217 |
| beta blocker, n (%) | 4 (23.5) | 32 (15.3) | 0.373 |
| cilostazol, n (%) | 4 (23.5) | 45 (21.5) | 0.847 |
| Simplified cough test abnormal, n (%) | 10 (58.8) | 8 (3.8) | <0.001 |

SD, standard deviation; ATBI, atherothrombotic brain infarction; CEI, cardiogenic embolism infarction; LI, lacunar infarction; ICH, intracerebral hemorrhage; NIHSS, National Institutes of Health Stroke Scale; RSST, repetitive saliva swallowing test; MWST, modified water swallowing test.

Table 4. The sensitivities of these tests, including the simplified cough test, were not high enough to make them useful as screening tests for predicting the pneumonia risk. Combining these tests raised the sensitivities, whereby an abnormal result on any one of the three tests showed a sensitivity of 88.2% and specificity of 83.7%.

**Table 3. Factors influencing pneumonia onset.**

| Factor | Model 1 | | | Model 2 | | | Model 3 | | |
|---|---|---|---|---|---|---|---|---|---|
| | Odds ratio | 95% CI | p value | Odds ratio | 95% CI | p value | Odds ratio | 95% CI | p value |
| NIHSS score | 1.16 | 1.04–1.31 | 0.010 | 1.23 | 1.11–1.35 | <0.001 | 1.23 | 1.10–1.38 | <0.001 |
| RSST < 3 times/30 seconds | 4.51 | 1.12–18.40 | 0.034 | | | | | | |
| MWST score < 4 | | | | 2.92 | 0.68–12.66 | 0.167 | | | |
| Abnormal simplified cough test result | | | | | | | 24.86 | 6.60–93.65 | <0.001 |

CI, confidence interval; NIHSS, National Institutes of Health Stroke Scale; RSST, repetitive saliva swallowing test; MWST, modified water swallowing test.

## Discussion

In this study, we evaluated the simplified cough test and found that an abnormal simplified cough test result represented an independent risk factor for aspiration pneumonia. In addition, an abnormal simplified cough test was associated with an abnormal RSST, which suggested that silent aspiration might influence swallowing dysfunction.

The utility of the cough test, in which patients orally inhale a mist of citric acid–physiological saline using a nebulizer for detecting silent aspiration, has previously been reported [14]. The utility was established by comparing the cough test with videofluoroscopic/videoendoscopic examinations, which are the gold standard evaluations. A previous systematic review of methods of citric acid cough reflex testing employed in 129 studies found that instrumentation and protocols differed widely across studies [23]. As mentioned earlier, in the original method, evoking up to 5 coughs within 1 minute was judged as normal; however, this test markedly strained patients. Therefore, Sato *et al.* [15] modified the test to judge the cough reflex within

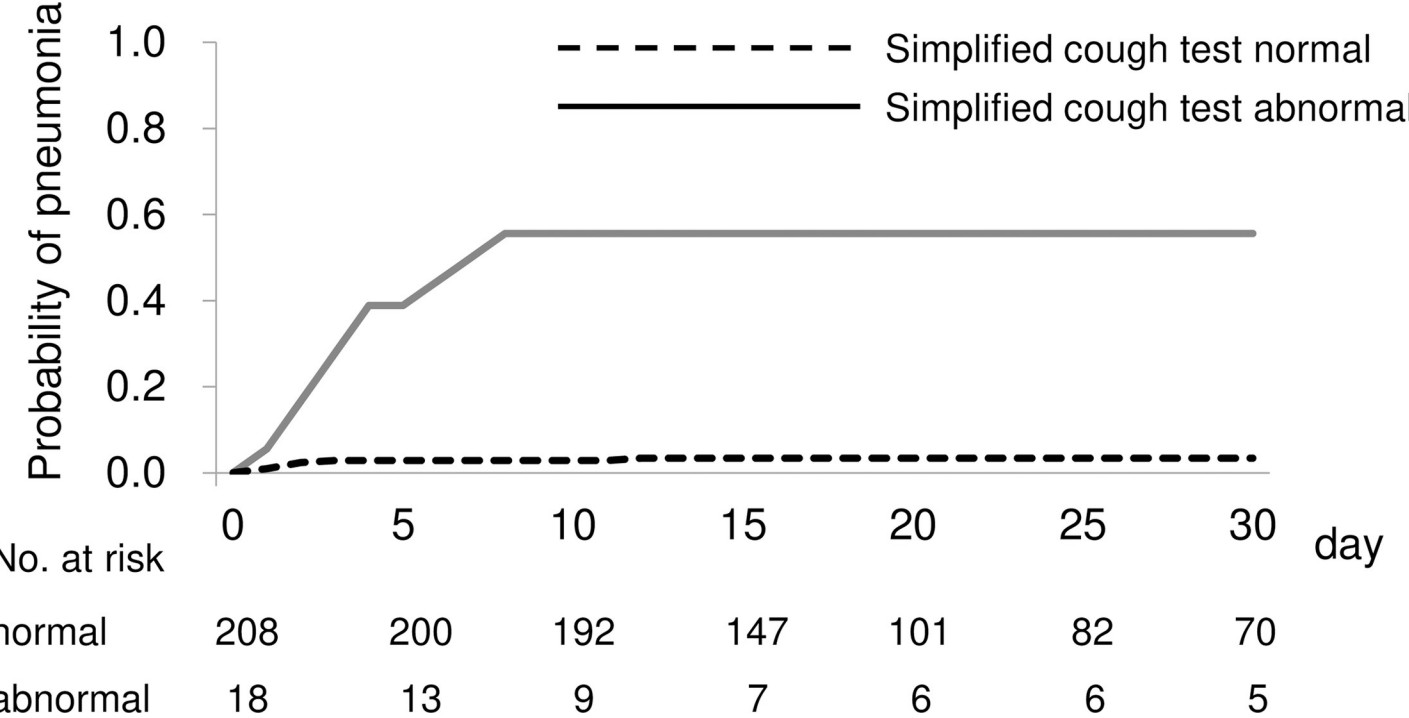

No. at risk

| | 0 | 5 | 10 | 15 | 20 | 25 | 30 |
|---|---|---|---|---|---|---|---|
| normal | 208 | 200 | 192 | 147 | 101 | 82 | 70 |
| abnormal | 18 | 13 | 9 | 7 | 6 | 6 | 5 |

**Fig 1. Relationship between simplified cough test results and pneumonia.** Kaplan–Meier curves of the duration until pneumonia development between patients with normal and abnormal simplified cough test results. The patient group with abnormal simplified cough test results had a higher incidence of pneumonia.

**Table 4. Predictive values of the bedside tests for pneumonia.**

|  | Sensitivity (%) | Specificity (%) | Positive predictive value (%) | Negative predictive value (%) | Accuracy (%) |
|---|---|---|---|---|---|
| RSST < 3 times/30 seconds | 64.70 | 87.6 | 29.700 | 96.80 | 85.8 |
| MWST score < 4 | 29.40 | 95.2 | 33.300 | 94.30 | 90.3 |
| Abnormal simplified cough test result | 58.80 | 96.2 | 55.600 | 96.60 | 93.4 |

RSST, repetitive saliva swallowing test; MWST, modified water swallowing test.

30 seconds, and found that coughing within 30 seconds was a threshold for detecting silent aspiration in aspirator patients (sensitivity, 92%; specificity, 94%). The usefulness of this simplified cough test has been validated by Lee *et al.* [16], who measured the time to the first cough as the latency and found a threshold of 28.12 seconds for predicting silent aspiration in an aspirator cohort (sensitivity, 87.1%; specificity, 66.7%). Moreover, Lee *et al.* reported that a cut-off value of 28.12 seconds rather than 60 seconds was sufficient to predict silent aspiration in a dysphagic patient cohort (sensitivity, 87.1%; specificity, 70.0%) as well as in an aspirator patient cohort. These reports suggest that a cut-off time of about 30 seconds is the most suitable for detecting silent aspiration. Thus, the simplified cough test is an appropriate screening test, as it can be rapidly performed while the patient is in a supine position, and with minimal patient cooperation.

Here, we compared the predictive values of several bedside tests for pneumonia in patients with acute stroke. Among the RSST, MWST, and simplified cough test, none had a sensitivity high enough to make the test useful as a screening test for the risk of pneumonia in this patient population. However, combining these three tests raised the sensitivity to 88.2%. Among our 226 patients, 17 patients developed pneumonia, of whom 10 patients had an abnormal simplified cough test result. Among these 10 patients, 6 patients had a normal RSST or MWST test result, which revealed that conventional bedside screening tests cannot predict the risk of aspiration pneumonia adequately. These findings suggest that the simplified cough test should be added to the existing bedside screening tests.

Our results are in line with those of a prior report demonstrating that the simplified cough test does not seem to be a useful standalone tool to screen for silent aspiration in patients with subacute stroke and suspected dysphagia [24]. In the present study, the subjects were patients with acute-phase stroke; in this phase, many patients do not have the protective cough reflex. In fact, all patients in our study who were diagnosed with pneumonia and had an abnormal simplified cough test result developed pneumonia within 10 days of admission. The simplified cough test is therefore more useful in the acute phase of stroke than in the subacute or chronic phase. Moreover, in this study, the endpoint was defined as pneumonia onset. In many studies, silent aspiration was evaluated by videofluoroscopic examination. However, this test may not reflect the actual situation. Thus, to predict pneumonia onset, the simplified cough test may be a useful tool. Further repeated and time-series examinations that include both the simplified cough test and videofluoroscopic evaluation should be performed in the future.

In this study, the association between medications (angiotensin-converting enzyme inhibitor, beta blocker, and cilostazol) and the result of simplified cough test or pneumonia onset was not detected. It has been reported that angiotensin-converting enzyme inhibitor promotes cough reflex and prevents pneumonia [19]. Beta blocker promotes bronchospasm and cough [20]. In addition, cilostazol which is an antiplatelet drug and inhibits phosphodiesterase, evokes cough [21]. In this study, we evaluated the patients in acute phase of stroke and the study period was short. Selection bias and the period of evaluation might influence on these results.

This study had several limitations. First, the study was conducted in a single institution, and thus may be biased by the single-center effect and clustering of observations. In fact, the median NIHSS score and the frequency of abnormal cough test were relatively low. One of the reasons was that the severe stroke patients were excluded with the criteria of this study. Sampling bias may also exist. Future multicenter collaborative research is needed to eliminate such effects. Second, the simplified cough test is only one component of the diagnosis of swallowing-related disorder. In this study, we could not compare the simplified cough test with the other instrumental swallowing examinations. However, there are previous reports which revealed the relationship between the simplified cough test and videofluoroscopic examination [12, 13, 15, 16]. While videofluoroscopic or videoendoscopic evaluations of swallowing have been used as the gold standard for evaluating aspiration, these methods have limitations, such as exposure to radiation and difficulties in conducting the examinations in disabled patients [25]. Additionally, it is not possible to perform videofluoroscopic examinations for all patients with acute stroke. Hence, it is valuable to be able to evaluate the risk of swallowing dysfunction and aspiration by using a combination of simple modalities at the bedside [9].

In this study, we showed that the simplified cough test is useful for predicting aspiration pneumonia. On the other hand, it is also important to develop a management in prevention of aspiration pneumonia. If the simplified cough test is abnormal, the risk of silent aspiration is high. Then, the patients should be carefully monitored the state of respiration such as peripheral capillary oxygen saturation. Previous reports emphasized the effective management protocol based on the results of cough reflex test [26, 27]. Miles *et al.* could not show improvement of pneumonia and mortality using the cough reflex test [26]. Conversely, Perry *et al.* made a strict flow chart for dysphagia and silent aspiration, which resulted in decreasing the frequency of pneumonia [27]. Also, in the point of rehabilitation, voluntary cough skill training is useful to improve the cough reflex in Parkinson's disease rehabilitation [28]. In addition, the improvement of sensitivity of respiratory tract should be considered. There are several methods to enhance the strength of swallowing related muscle, whereas there are few methods to enhance the sensory function. Recently, the electrical therapeutic machine to stimulate the sensory nerve of pharyngeal and laryngeal mucosa has been developed [29]. Such an instrument will be expected to prevent silent aspiration and pneumonia. Further trial should be performed.

## Conclusions

The simplified cough test is a strong indicator for predicting pneumonia development in patients with acute stroke. Therefore, the simplified cough test should be added to the existing bedside screening tests to gauge the risk of pneumonia, allowing appropriate and timely intervention.

## Supporting information

**S1 Data. All relevant data of the study.**
(XLSX)

## Acknowledgments

We would like to sincerely thank the staff at Suiseikai Kajikawa Hospital for their technical assistance.

## Author Contributions

**Conceptualization:** Masahiro Nakamori, Eiji Imamura, Miyu Kuwabara, Tomoko Ayukawa, Yuki Hayashi, Hayato Matsushima, Mika Funai.

**Data curation:** Masahiro Nakamori, Eiji Imamura, Miyu Kuwabara, Tomoko Ayukawa, Keisuke Tachiyama, Teppei Kamimura, Yuki Hayashi.

**Formal analysis:** Masahiro Nakamori.

**Investigation:** Masahiro Nakamori, Eiji Imamura, Miyu Kuwabara, Tomoko Ayukawa, Keisuke Tachiyama, Teppei Kamimura, Yuki Hayashi, Hayato Matsushima, Mika Funai.

**Methodology:** Masahiro Nakamori, Eiji Imamura, Miyu Kuwabara, Tomoko Ayukawa, Yuki Hayashi, Mika Funai.

**Project administration:** Masahiro Nakamori.

**Resources:** Masahiro Nakamori, Miyu Kuwabara.

**Supervision:** Tatsuya Mizoue, Shinichi Wakabayashi.

**Validation:** Masahiro Nakamori, Eiji Imamura, Yuki Hayashi, Mika Funai, Tatsuya Mizoue, Shinichi Wakabayashi.

**Writing – original draft:** Masahiro Nakamori, Eiji Imamura, Yuki Hayashi, Tatsuya Mizoue, Shinichi Wakabayashi.

**Writing – review & editing:** Masahiro Nakamori, Eiji Imamura, Tatsuya Mizoue, Shinichi Wakabayashi.

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
