## [Decision Letter · Decision Letter 0]

18 Aug 2020

PONE-D-20-20611

Simplified cough test can predict the risk for pneumonia in patients with acute stroke

PLOS ONE

Dear Dr. Nakamori,

Thank you for submitting your manuscript to PLOS ONE. After careful consideration, we feel that it has merit but does not fully meet PLOS ONE’s publication criteria as it currently stands. Therefore, we invite you to submit a revised version of the manuscript that addresses the points raised during the review process.

ACADEMIC EDITOR: I have received the comments of the reviewers on your manuscript. The specific comments of the reviewers are included below. Please provide point by point response in your revised manuscript.

We look forward to receiving your revised manuscript.

Kind regards,

Muhammad Adrish

Academic Editor

PLOS ONE

Journal Requirements:

2.Thank you for stating the following financial disclosure:

 [N/A].

3.Thank you for stating the following in your Competing Interests section: 

[N/A].

Reviewers' comments:

Reviewer's Responses to Questions

**Comments to the Author**

1. Is the manuscript technically sound, and do the data support the conclusions?

Reviewer #1: Yes

Reviewer #2: Partly

2. Has the statistical analysis been performed appropriately and rigorously? 

Reviewer #1: Yes

Reviewer #2: Yes

3. Have the authors made all data underlying the findings in their manuscript fully available?

Reviewer #1: Yes

Reviewer #2: Yes

4. Is the manuscript presented in an intelligible fashion and written in standard English?

Reviewer #1: Yes

Reviewer #2: Yes

5. Review Comments to the Author

Reviewer #1: Aspiration pneumonia is a critical issue in patients with stroke. The authors investigated the association

between the results of a simplified cough test, aspiration risk and pneumonia onset in

226 patients with acute stroke. No patient had pneumonia before stroke onset.

For the simplified cough test, performed on admission, patients orally inhaled a mist of 1% citric acid–

physiological saline using a portable mesh nebulizer. When the first cough was evoked or if it remained

absent for 30 seconds (indicating an abnormal result), the test was ended. Patients also completed the

repetitive saliva swallowing test (RSST) and modified water swallowing test. Patients were monitored for

pneumonia signs for 30 days following admission. Eighteen patients exhibited an abnormal simplified cough

test result. On multivariate analysis, an abnormal RSST result was independently associated with an

abnormal simplified cough test result. Seventeen patients developed pneumonia.

Statistical analysis revealed that the simplified cough test had predictive

power for pneumonia onset (hazard ratio, 10.52). Seventeen patients developed pneumonia, of whom 10 patients had an abnormal simplified cough test result. Among these 10 patients, 6 patients had a normal RSST or MWST test result, which revealed that conventional bedside screening tests cannot predict the risk of aspiration pneumonia adequately. These findings suggested that the simplified cough test should be added to the existing bedside screening tests.

The authors concluded that the simplified cough test is a strong indicator for predicting

the pneumonia development in patients with acute stroke and that it should be added to existing

bedside screening tests for predicting pneumonia risk, allowing

appropriate and timely intervention. In addition, an abnormal simplified cough test

was associated with an abnormal RSST, which suggested that silent aspiration might

influence swallowing dysfunction.

General comments:

This is an interesting study. Mainly, it describes a simple, noninvasive and inexpensive means of assessing

risk for aspiration and pneumonia in patients with acute stroke as an alternative to videoesophagofluoroscopy/endoscopy.

The paper is well-constructed, and contains the necessary sections of introduction, question asked, methods, results, discussion and conclusions.

Specific comments:

1. Since patients were admitted with acute stroke, I would assume many, if not all, had hypertension and were being treated for it. Were any patients receiving beta-blockers or ACE inhibitors?

Both classes of medications are known to cause bronchospasm and/or cough which potentially could make the airways more reactive and lower threshold for cough.

2. Were patients with neuromuscular disorders also excluded? They often have bulbar dysfunction and may be hard to distinguish from strokes with brain stem involvement.

3. Tables should be placed at the end of the paper, rather than amongst the text.

4. Line 149: Small point, but in Table 1, the age of all 226 patients is said to be 72.8 ± 13.1, but the age of the 208 “normal” subjects was 74.2 ± 15.9. Since this is the larger group by far, I would expect the overall mean to be closer to 74 rather than to 72.8. Please check the calculations. In all the other variables, the overall mean is closer to the “normal” means.

Reviewer #2: This paper is very interesting because it focuses on a convenient method available at the bedside. It will be more valuable with some modifications.

1, Please clarify the concrete method how to prepare the orally inhaled a mist of 1% citric acid. Is it made of medical drug or general product?

2, Was the NIHSS measured only once? Or was it measured repeatedly? Please tell me how many times the authors evaluated on average per patients.

3, I think it is better to clarify the interventions of the stroke to each patient.

4, If the simplified cough test is positive, what should clinicians be careful of in daily medical care? Is there any specific methods to prevent aspiration pneumonia? Please suggest such contents in discussion.

6. PLOS authors have the option to publish the peer review history of their article (what does this mean?). If published, this will include your full peer review and any attached files.

Reviewer #1: No

Reviewer #2: No

---

## [Author Response · Author response to Decision Letter 0]

26 Aug 2020

We appreciate your advice. The manuscript has been revised as follows.

Response to the editor

Comment 1: Please ensure that your manuscript meets PLOS ONE's style requirements, including those for file naming. The PLOS ONE style templates can be found at https://journals.plos.org/plosone/s/file?id=wjVg/PLOSOne_formatting_sample_main_body.pdf and https://journals.plos.org/plosone/s/file?id=ba62/PLOSOne_formatting_sample_title_authors_affiliations.pdf

Response 1: We have ensured that the manuscript meets PLOS ONE's style requirements.

Comment 2: Thank you for stating the following financial disclosure: [N/A]. At this time, please address the following queries:

a. Please clarify the sources of funding (financial or material support) for your study. List the grants or organizations that supported your study, including funding received from your institution.

d. If you did not receive any funding for this study, please state: “The authors received no specific funding for this work.”

Response 2: We have added the following sentence in the Funding section and included the statement in the cover letter:

The authors received no specific funding for this work.

Comment 3: Thank you for stating the following in your Competing Interests section: [N/A]. Please complete your Competing Interests on the online submission form to state any Competing Interests. If you have no competing interests, please state "The authors have declared that no competing interests exist.", as detailed online in our guide for authors at http://journals.plos.org/plosone/s/submit-now

Response 3: We have added the following sentence in the Competing Interests section and included this statement in the cover letter:

The authors have declared that no competing interests exist.

Response to the reviewers

Reviewer #1: Aspiration pneumonia is a critical issue in patients with stroke. The authors investigated the association between the results of a simplified cough test, aspiration risk and pneumonia onset in 226 patients with acute stroke. No patient had pneumonia before stroke onset.

For the simplified cough test, performed on admission, patients orally inhaled a mist of 1% citric acid–physiological saline using a portable mesh nebulizer. When the first cough was evoked or if it remained absent for 30 seconds (indicating an abnormal result), the test was ended. Patients also completed the repetitive saliva swallowing test (RSST) and modified water swallowing test. Patients were monitored for pneumonia signs for 30 days following admission. Eighteen patients exhibited an abnormal simplified cough test result. On multivariate analysis, an abnormal RSST result was independently associated with an abnormal simplified cough test result. Seventeen patients developed pneumonia. Statistical analysis revealed that the simplified cough test had predictive power for pneumonia onset (hazard ratio, 10.52). Seventeen patients developed pneumonia, of whom 10 patients had an abnormal simplified cough test result. Among these 10 patients, 6 patients had a normal RSST or MWST test result, which revealed that conventional bedside screening tests cannot predict the risk of aspiration pneumonia adequately. These findings suggested that the simplified cough test should be added to the existing bedside screening tests. The authors concluded that the simplified cough test is a strong indicator for predicting the pneumonia development in patients with acute stroke and that it should be added to existing bedside screening tests for predicting pneumonia risk, allowing appropriate and timely intervention. In addition, an abnormal simplified cough test was associated with an abnormal RSST, which suggested that silent aspiration might influence swallowing dysfunction. 

General comments:

This is an interesting study. Mainly, it describes a simple, noninvasive and inexpensive means of assessing risk for aspiration and pneumonia in patients with acute stroke as an alternative to videoesophagofluoroscopy/endoscopy. The paper is well-constructed, and contains the necessary sections of introduction, question asked, methods, results, discussion and conclusions.

Specific comments:

Comment 1: Since patients were admitted with acute stroke, I would assume many, if not all, had hypertension and were being treated for it. Were any patients receiving beta-blockers or ACE inhibitors? Both classes of medications are known to cause bronchospasm and/or cough which potentially could make the airways more reactive and lower threshold for cough.

Response 1: We appreciate this important suggestion. Some drugs tend to promote cough reflex. We analyzed the usage of ACE inhibitors and beta-blockers as well as cilostazol. Then, we statistically analyzed their influence on the result of the simplified cough test. In this study, we could not find an association between these drugs and the simplified cough test or pneumonia onset. We considered this was possibly due to the fact that these evaluations were performed in the acute phase of stroke. We added sentences in Methods, Results, and Discussion sections as follows.

Page 6, Lines 121-123, 

In addition, the medications of angiotensin-converting enzyme inhibitor, beta blocker, and cilostazol (antiplatelet drug inhibiting phosphodiesterase), which promote cough reflex were investigated [19-21].

Page 7, Lines 160-162,

The medications of angiotensin-converting enzyme inhibitor, beta blocker, and cilostazol were not associated with the results of simplified cough test.

Page 11, Lines 178-179,

The medications of angiotensin-converting enzyme inhibitor, beta blocker, and cilostazol were not associated with the results of pneumonia onset.

Pages 16-17, Lines 262-269,

In this study, the association between medications (angiotensin-converting enzyme inhibitor, beta blocker, and cilostazol) and the result of simplified cough test or pneumonia onset was not detected. It has been reported that angiotensin-converting enzyme inhibitor promotes cough reflex and prevents the pneumonia [19]. Beta blocker promotes bronchospasm and cough [20]. In addition, cilostazol which is an antiplatelet drug and inhibits phosphodiesterase, evokes cough [21]. In this study, we evaluated the patients in acute phase of stroke and the study period was short. Selection bias and the period of evaluation might influence on these results.

19. Sekizawa K, Matsui T, Nakagawa T, Nakayama K, Sasaki H (1998) ACE inhibitors and 

pneumonia. Lancet 352: 1069.

20. Uzubalis RI, Frewin DB, Bushell MK, McEvoy RD (1989) The effect of angiotensin converting 

enzyme inhibitors (ACE-I) and selective beta 1-antagonists on bronchial reactivity and the 

cough reflex in man. Eur J Clin Pharmacol 37: 467-470.

21. Shinohara Y (2006) Antiplatelet cilostazol is effective in the prevention of pneumonia in 

ischemic stroke patients in the chronic stage. Cerebrovasc Dis 22: 57-60.

Comment 2: Were patients with neuromuscular disorders also excluded? They often have bulbar dysfunction and may be hard to distinguish from strokes with brain stem involvement.

Response 2: We excluded patients with neuromuscular disease. In addition, we also excluded the patients who had received an operation or endovascular treatment, because they had received anesthetic drugs or sedatives. We added sentences to the Methods section as follows.

Page 5, Lines 90-94,

We excluded patients who were not eating food before stroke onset, who were in a coma (best eye response score on the Glasgow coma scale of 1), who had received an operation or endovascular treatment, or who were on mechanical ventilation. Patients who had a history of neuromuscular disorders or asthma were also excluded.

Comment 3: Tables should be placed at the end of the paper, rather than amongst the text.

Response 3: According to the rules for submission to PLOS ONE, we must place the tables within the text. 

Comment 4: Line 149: Small point, but in Table 1, the age of all 226 patients is said to be 72.8 ± 13.1, but the age of the 208 “normal” subjects was 74.2 ± 15.9. Since this is the larger group by far, I would expect the overall mean to be closer to 74 rather than to 72.8. Please check the calculations. In all the other variables, the overall mean is closer to the “normal” means.

Response 4: We apologize that we wrote the reverse values of “normal” and “abnormal” patients. We revised and rechecked all other calculations and made the appropriate corrections. 

Reviewer #2: This paper is very interesting because it focuses on a convenient method available at the bedside. It will be more valuable with some modifications.

Comment 1: Please clarify the concrete method how to prepare the orally inhaled a mist of 1% citric acid. Is it made of medical drug or general product?

Response 1: We appreciate your suggestion. We rewrote the sentence. Citric acid is not a medical drug and was purchased from a pharmaceutical company. We added to the Methods section as follows.

Page 5, Lines 98-99,

Citric acid was purchased from Kenei Pharmaceutical Co., Ltd. (Osaka, Japan).

Comment 2: Was the NIHSS measured only once? Or was it measured repeatedly? Please tell me how many times the authors evaluated on average per patients.

Response 2: We appreciate your suggestion. In our institution, two stroke specialists examined the patients on admission and determined the NIHSS score. After admission, all patients were routinely evaluated again 3 hours, 6 hours, 12 hours, and 24 hours later by staff. In this article, we recorded the NIHSS score on admission at the same time of simplified cough test. We added in Methods section as follows.

Page 6, Line 117,

Two stroke specialists evaluated the stroke severity of the patients on admission.

Comment 3: I think it is better to clarify the interventions of the stroke to each patient.

Response 3: We appreciate your suggestion. In this study, patients who had operations performed including endovascular therapy were excluded because they had received anesthesia or sedation. Drugs such as beta-blockers, ACE inhibitors, and cilostazol tend to evoke coughs. Thus, we evaluated the usage of these drugs, and we statistically analyzed their influence on the results. We modified Tables 1 and 2 and added the following sentences to the Methods, Results, and Discussion sections as follows.

Page 5, Lines 90-94,

We excluded patients who were not eating food before stroke onset, who were in a coma (best eye response score on the Glasgow coma scale of 1), who received the operation or endovascular treatment or who were on mechanical ventilation. Patients who had a history of neuromuscular disorders and asthma were also excluded.

Page 6, Lines 121-123, 

In addition, the medications of angiotensin-converting enzyme inhibitor, beta blocker, and cilostazol (antiplatelet drug inhibiting phosphodiesterase), which promote cough reflex were investigated [19-21].

Page 7, Lines 160-162,

The medications of angiotensin-converting enzyme inhibitor, beta blocker, and cilostazol were not associated with the results of simplified cough test.

Page 10, Lines 177-178,

The medications of angiotensin-converting enzyme inhibitor, beta blocker, and cilostazol were not associated with the results of pneumonia onset.

Pages 16-17, Lines 262-269,

In this study, the association between medications (angiotensin-converting enzyme inhibitor, beta blocker, and cilostazol) and the result of simplified cough test or pneumonia onset was not detected. It has been reported that angiotensin-converting enzyme inhibitor promotes cough reflex and prevents the pneumonia [19]. Beta blocker promotes bronchospasm and cough [20]. In addition, cilostazol which is an antiplatelet drug and inhibits phosphodiesterase, evokes cough [21]. In this study, we evaluated the patients in acute phase of stroke and the study period was short. Selection bias and the period of evaluation might influence on these results.

19. Sekizawa K, Matsui T, Nakagawa T, Nakayama K, Sasaki H (1998) ACE inhibitors and 

pneumonia. Lancet 352: 1069.

20. Uzubalis RI, Frewin DB, Bushell MK, McEvoy RD (1989) The effect of angiotensin converting 

enzyme inhibitors (ACE-I) and selective beta 1-antagonists on bronchial reactivity and the 

cough reflex in man. Eur J Clin Pharmacol 37: 467-470.

21. Shinohara Y (2006) Antiplatelet cilostazol is effective in the prevention of pneumonia in 

ischemic stroke patients in the chronic stage. Cerebrovasc Dis 22: 57-60.

Comment 4: If the simplified cough test is positive, what should clinicians be careful of in daily medical care? Is there any specific methods to prevent aspiration pneumonia? Please suggest such contents in discussion.

Response 4: We appreciate your suggestion. If the simplified cough test is abnormal, the risk of silent aspiration is high. Then, patients should be carefully monitored as to the state of respiration, such as peripheral capillary oxygen saturation. In addition, the improvement of sensitivity of respiratory tract should be considered. There are several methods to enhance the strength of swallowing related muscles, whereas there are few methods to enhance sensory function. Recently, an electrical therapeutic machine to stimulate the sensory nerve of pharyngeal and laryngeal mucosa has been developed. Such an instrument will be expected to prevent silent aspiration and pneumonia. Further trials should be performed. We added these discussion as follows.

Page 17, Lines 286-301

In this study, we showed that the simplified cough test is useful for predicting aspiration pneumonia. On the other hand, it is also important to develop a management in prevention of aspiration pneumonia. If the simplified cough test is abnormal, the risk of silent aspiration is high. Then, the patients should be carefully monitored the state of respiration such as peripheral capillary oxygen saturation. Previous reports emphasized the effective management protocol based on the results of cough reflex test [26,27]. Miles et al. could not show improvement of pneumonia and mortality using the cough reflex test [26]. Conversely, Perry et al. made a strict flow chart for dysphagia and silent aspiration, which resulted in decreasing the frequency of pneumonia [27]. Also, in the point of rehabilitation,voluntary cough skill training is useful to improve the cough reflex in Parkinson’s disease rehabilitation [28]. In addition, the improvement of sensitivity of respiratory tract should be considered. There are several methods to enhance the strength of swallowing related muscle, whereas there are few methods to enhance the sensory function. Recently, the electrical therapeutic machine to stimulate the sensory nerve of pharyngeal and laryngeal mucosa has been developed [29]. Such an instrument will be expected to prevent silent aspiration and pneumonia. Further trial should be performed.

26. Miles A, Zeng IS, McLauchlan H, Huckabee ML (2013) Cough reflex testing in Dysphagia 

following stroke: a randomized controlled trial. J Clin Med Res 5: 222-233.

27. Perry SE, Miles A, Fink JN, Huckabee ML (2019) The Dysphagia in Stroke Protocol Reduces 

Aspiration Pneumonia in Patients with Dysphagia Following Acute Stroke: a Clinical Audit. 

Transl Stroke Res 10: 36-43.

28. Curtis JA, Dakin AE, Troche MS (2020) Respiratory-Swallow Coordination Training and 

Voluntary Cough Skill Training: A Single-Subject Treatment Study in a Person With 

Parkinson's Disease. J Speech Lang Hear Res 63: 472-486.

29. Nagami S, Maeda K, Fukunaga S, Ikeno M, Oku Y (2019) Safety of transcutaneous electrical 

sensory stimulation of the neck in terms of vital parameters in dysphagia rehabilitation. Sci Rep 

9: 13481.

---

## [Decision Letter · Decision Letter 1]

10 Sep 2020

Simplified cough test can predict the risk for pneumonia in patients with acute stroke

PONE-D-20-20611R1

Dear Dr. Nakamori,

We’re pleased to inform you that your manuscript has been judged scientifically suitable for publication and will be formally accepted for publication once it meets all outstanding technical requirements.

Kind regards,

Muhammad Adrish

Academic Editor

PLOS ONE

Additional Editor Comments (optional):

You have satisfactorily answered all queries raised by the reviewers.

Reviewers' comments:

Reviewer's Responses to Questions

**Comments to the Author**

1. If the authors have adequately addressed your comments raised in a previous round of review and you feel that this manuscript is now acceptable for publication, you may indicate that here to bypass the “Comments to the Author” section, enter your conflict of interest statement in the “Confidential to Editor” section, and submit your "Accept" recommendation.

Reviewer #1: All comments have been addressed

Reviewer #2: All comments have been addressed

2. Is the manuscript technically sound, and do the data support the conclusions?

Reviewer #1: Yes

Reviewer #2: Yes

3. Has the statistical analysis been performed appropriately and rigorously? 

Reviewer #1: Yes

Reviewer #2: Yes

4. Have the authors made all data underlying the findings in their manuscript fully available?

Reviewer #1: Yes

Reviewer #2: Yes

5. Is the manuscript presented in an intelligible fashion and written in standard English?

Reviewer #1: Yes

Reviewer #2: Yes

6. Review Comments to the Author

Reviewer #1: Reviewers' comments have been addressed. Paper is more clear and precise now.

Reviewer #2: (No Response)

7. PLOS authors have the option to publish the peer review history of their article (what does this mean?). If published, this will include your full peer review and any attached files.

Reviewer #1: No

Reviewer #2: No

---

## [Editor Report · Acceptance letter]

14 Sep 2020

PONE-D-20-20611R1 

Simplified cough test can predict the risk for pneumonia in patients with acute stroke 

Dear Dr. Nakamori:

I'm pleased to inform you that your manuscript has been deemed suitable for publication in PLOS ONE. Congratulations! Your manuscript is now with our production department. 

Kind regards, 

on behalf of

Dr. Muhammad Adrish 

Academic Editor

PLOS ONE